# Differences in the fitness effects of traded resources shape traits and persistence in multi-mutualist communities

Renuka Agarwal[1]\*, Anne E. Curé[1], Kari A. Segraves[2], David M. Althoff[1]\*

**1** Department of Biology, Syracuse University, College Place, Syracuse, New York, United States of America, **2** National Science Foundation, Eisenhower Ave, Alexandria, Virginia, United States of America

\* renukaag2011@gmail.com (RA); dmalthof@syr.edu (DMA)

## Abstract

Mutualistic interactions, where species reciprocally benefit from each other, are crucial for ecosystem stability and biodiversity. These interactions often involve species that experience different fitness effects for the traded resources or services. Because mutualisms rely on positive feedback between partners, such asymmetries can strongly influence evolutionary outcomes. Differences in fitness effects create divergent selective pressures, shaping trait evolution and determining the persistence of mutualisms. The strength of these effects can also vary depending on the availability of traded resources from other sources. Despite their importance, the evolutionary role of fitness asymmetries in mutualism has received little attention, beyond recognizing that some species may be more dependent on their partners than others. This study investigates how asymmetry in the fitness effects of a traded resource influences the persistence and phenotypic trait evolution of species in multi-mutualist guilds. To test this, we constructed synthetic multi-mutualist communities by combining, reproductively isolated and genetically modified strains of *Saccharomyces cerevisiae* that engage in a nutritional mutualism by trading adenine and lysine. One guild of four strains cannot produce lysine but overproduces adenine while the other guild cannot produce adenine but overproduces lysine. Lysine overproducers survive periods of low adenine better than adenine overproducers survive low lysine. Over a four-week evolution experiment we observed that strain persistence was strongly influenced by the availability of external resources. Communities in media containing traded resources supported the survival of all strains, whereas obligate conditions led to a significant extinction, especially for adenine overproducers. We observed distinct evolutionary trajectories of traits under obligate versus supplemented conditions. Phenotypic assays revealed that costs and benefits evolved differently depending on the essentiality of the traded resource and nutrient supplementation. These results demonstrate that asymmetries in the fitness effects of traded resources can influence evolutionary outcomes, species persistence, and community stability in multi-mutualist communities.

**Data availability statement:** The data underlying the results presented in this study are publicly available through Dryad at the following DOI: https://doi.org/10.5061/dryad.kwh70rzjq.

**Funding:** Grants from the U.S. National Science Foundation (DEB 2137554). The funders had no role in study design, data collection and analysis, decision to publish, or the preparation of the manuscript.

**Competing interests:** The authors have declared that no competing interests exist.

## Introduction

Species engage in a range of interactions, from parasitism to competition to mutualism [1]. Among these, mutualism stands out as reciprocally beneficial interaction in which two or more partners gain fitness benefits [2]. Such relationships are essential for the stability and functioning of ecosystems, as they involve exchange of essential resources that contribute to the foundation of many communities [3–5]. Mutualisms can also expand to involve a network of interactions with multiple partner species [6–7]. From the well-known associations between plants and pollinators to complex symbioses between microbes and their hosts, mutualism facilitates key processes such as nutrient cycling, pollination, and defense against predators [2]. Beyond their ecological importance, mutualistic interactions provide valuable insights into evolutionary biology, particularly in understanding how reciprocity evolves and is maintained among species [8–10]. Mutualistic interactions have been studied by focusing on both individual pairs of interacting species as well as guilds or networks of potential mutualistic partner species [2]. Mutualisms are highly diverse, occurring across a wide range of environments and are influenced by both abiotic and biotic factors. For instance, the coexistence of phylogenetically diverse bacteria within a single coral [11], the co-occurrence of multiple endosymbionts in the insect guts [12], and the diversity of arbuscular mycorrhizal fungi in the roots of plants [13]. As most mutualistic interactions in nature are multi-partner, it is highly important to understand the complex interplay between ecological and evolutionary dynamics that govern these multi-species mutualisms.

The direct and indirect interaction feedbacks that can occur in multi-mutualist communities [14] are influenced by the availability of external sources of commodities essential for mutualistic interactions, as well as by changes in species richness driven by shifts in the abiotic or biotic environment [15]. The availability of alternative resources can change the intensity of selection pressures, thereby modifying the degree of dependency and number of partners within mutualistic communities. Both these factors can shape the evolution of mutualistic traits [16]. As alternative sources of traded resources or services become more available, partners can persist to some extent independently, and the benefits of interactions become more flexible [17]. In contrast, extreme dependency in mutualistic relationships can drive specialization, where the survival of one mutualist species is solely dependent on its partner. In such cases, stronger selection pressures on mutualistic traits are expected [18].

The intensity of selection on mutualists may be influenced by the relative contribution of traded resources to each partner's fitness [17]. This concept is analogous to the life-dinner principle in predator-prey interactions, where the fitness cost to prey is much higher than to predators [19]. For example, in pollination mutualisms, if a pollinator does not visit a flower, it only misses a single nectar meal, whereas the plant loses the opportunity to fertilize ovules. Similarly, endosymbionts that fail to locate a host may perish, while long-lived host can survive without endosymbionts until colonized. Such asymmetries in the importance of mutualistic resources are expected to generate differential selective pressures on partners and may alter the rate of trait evolution. Asymmetries may also magnify competition for traded resources among

members of mutualistic guild, adding further selective pressures as mutualists adapt both to their partners and to guild members competing for the same resources [20]. Testing these effects of asymmetry is a next important step in closing a major gap in our understanding of mutualism [16].

The complexity of multi-mutualist communities and fitness asymmetries among partner species raises the question of how do asymmetries in the fitness consequences of traded resources change mutualism dynamics? This question is central to understanding the long-term stability and evolutionary trajectories of mutualistic systems, particularly in the context of environmental change [21–23]. To address this question, we used a synthetic nutritional mutualistic system based on brewer's yeast, *Saccharomyces cerevisiae,* in which we can modify the availability of the traded mutualistic resources, adenine and lysine. These nutritional resources differ in their relative importance for yeast survival and reproduction. Specifically, the importance of lysine to yeast cells is greater than for adenine [24–25], creating an inherent asymmetry in partner benefits. We assembled communities consisting of four reproductively isolated yeast strains on each side of the mutualism and maintained them for four weeks. We surveyed mutualistic strain persistence and key traits associated with production of traded resources and population growth.

## Materials and methods

### Synthetic model of multi-mutualist communities

We used genetically modified strains of *S. cerevisiae* [25] to study how the availability of traded resources and fitness asymmetries influences the persistence of mutualistic strains and the evolution of their phenotypic traits. The strains were derived from adenine- and lysine-overproducing parental strains WY833 and WY811, respectively [25]. To generate the lysine overproducing mutualists (LysOP), WY811 was crossed with SY9913 (Euroscarf 40007), and selected progeny were subsequently crossed to introduce pairwise combinations of *ura3Δ*, *his3Δ*, and *leu2Δ* auxotrophies used for selective tracking in co-culture. Similarly, the adenine overproducing mutualists (AdeOP) were engineered by crossing WY833 with a progenitor strain derived from WY833 × SY9913 crosses (genotype *ade8Δ0 his3Δ1 leu2Δ0 ura3Δ0 lys21WT*), followed by additional backcrossing to obtain the desired auxotrophic marker combinations. For a subset of strains, *trp1Δ* was introduced using PCR-based gene disruption with SY9914 (Euroscarf 40008) as a template to enable positive selection on hygromycin plates. All strains share a common laboratory genetic background and differ only at the engineered biosynthetic and auxotrophic loci, ensuring genetic comparability across treatments. These strains are genetically distinct, reproductively isolated, and functionally analogous to species. They engage in a cross-feeding nutritional mutualism, where each strain overproduces a resource necessary for the survival of its mutualist partner. The two types of overproducing guilds were adenine overproducers (AdeOP) that produce excess amounts of adenine but cannot produce their own lysine and lysine overproducers (LysOP) that produce excess amounts of lysine but cannot produce their own adenine. AdeOP strains release adenine immediately into the media, whereas the LysOP strains release a small amount of lysine while alive but release large amounts of lysine following cell death and lysis [24,26].

Adenine and lysine are essential nutrients for yeast cells, but they differ in their physiological effects when limited. Adenine is actively transported into yeast cells and incorporated into ribonucleotides [27]. A shortage of adenine prevents cell division within about two hours when ATP, ADP, and AMP have been mostly consumed [28]. During this period, yeast also start accumulating storage carbohydrates like trehalose and glycogen and mount a strong stress response that leads to increased survival [29]. Thus, adenine starved cells can survive for long periods of time and have carbohydrates stores that can be mobilized when conditions improve. Lysine, on the other hand, is also actively transported into yeast cells but is used as a nitrogen source for protein production [30]. Yeast cells that are starved for lysine begin programmed cell death and can start dying within 24 h [31]. Additionally, lysine starvation can lead to reduced resource use efficiency of glucose which limits growth [32] and causes increases in the presence of reactive oxygen species within the cell that leads to cell damage [31]. Because of this difference in the importance of adenine and lysine to cell functioning, the AdeOP strains have reduced starvation resistance relative to the LysOP strains [24–25] and will experience increased oxidative stress

after lysine starvation. Together, these differences in resource release dynamics, starvation resistance, and physiological responses establish a strong asymmetry between the mutualistic partners, providing a powerful system to experimentally test how resource availability and fitness imbalances shape the persistence, and evolutionary trajectories of mutualistic interactions.

We used 16 genotypes of yeast strains in different combinations to create symmetrical multi-species communities of four AdeOP strains and four LysOP strains (S1 Table). The combination of genotypes allowed us to separate the strains via selective plating. As the AdeOPs and the LysOPs mutualists do not produce lysine and adenine, respectively, they form an obligate mutualism when cultured in media lacking these nutrients. We set up communities in two environments: one without external adenine and lysine, where persistence depended entirely on partner exchange and one supplemented with low levels of adenine ($4 \times 10^{-3}$ g/L) and lysine ($18 \times 10^{-3}$ g/L), allowing yeast to obtain nutrients both from the media and from partner exchange. With low nutrient supplementation, the mutualisms become less dependent, as strains can partially meet their nutritional needs for adenine and lysine from the environment.

### Setting up the communities and evolving the strains

We used experimental evolution to explore the evolutionary changes of mutualists in response to each other. To initiate the evolution experiment, we cultured single colonies overnight in liquid YPD (10 g/L yeast extract, 20 g/L peptone, and 20 g/L dextrose). Yeast strains were then rinsed with sterile water and diluted to a standard density of 0.0125 $OD_{600}$. Mutualistic communities were assembled by combining four AdeOPs and four LysOPs into 2 ml cultures, with a starting density of 0.1 $OD_{600}$ (0.0125 $OD_{600}$ of each interacting strain) for each community (Fig 1). To establish a mutualism in the absence of external resources, we propagated mutualistic cultures in synthetic dextrose (SD) medium lacking adenine and lysine (1.5 g/L Difco yeast nitrogen base without amino acids or ammonium sulfate, 5 g/L ammonium sulfate, and 20 g/L dextrose, with additional amino acids). To establish a mutualism in the presence of external nutrients, the communities were grown in SD medium with 0.1X adenine and 0.2X lysine (1.5 g/L Difco yeast nitrogen base without amino acids or ammonium sulfate, 5 g/L ammonium sulfate, and 20 g/L dextrose, $4 \times 10^{-3}$ g/L adenine and $18 \times 10^{-4}$ g/L lysine and with supplemental amino acids), respectively, in a 48-well plate. We made 96 unique combinations of eight yeast strain communities. Each combination was replicated twice in media without external resources, resulting in a total of 192 independent cultures. For media containing low levels of adenine and lysine, there were 96 independent cultures. We chose this replication scheme because previous work [25] reported high rates of strain extinction under obligate conditions, and our goal was to maximize the number of communities that retained all eight strains. All cultures were incubated in their respective liquid medium on a rotating wheel in 48-well deep-well plates at 30°C. Cultures were grown for 48 hours in 2 ml of medium and typically reached a mean $OD_{600}$ of approximately 8 before transfer. At each transfer, 200 µl of the culture (10% of the total volume) was inoculated into 1.8 ml of fresh medium. This relatively high transfer volume ensured that the communities did not experience strong population bottlenecks and thus maintained a large effective population size throughout the experiment. As a result, selection was expected to act efficiently on phenotypic and genetic variation within the communities. The transfer regime did, however, did serve to reduce the number of cells per unit volume and did reduce the amount of traded resources right after a transfer event. At the end of each week, we checked for the presence of each strain by spot plating each community on eight selective plates that each only allowed growth for a single strain.

The experiment was run for four weeks. We froze the cultures that were used to initiate the experiment (designated as 'ancestral') and the cultures at the end of four weeks (designated as 'evolved'). The cultures were frozen in 25% glycerol and stored at −80°C.

### Assessing trait evolution

After four weeks of evolution, we assessed if any traits important to the mutualism had changed in the evolved strains compared to the ancestral strains. For evolved strains, we wanted to use strains which persisted in the most species rich

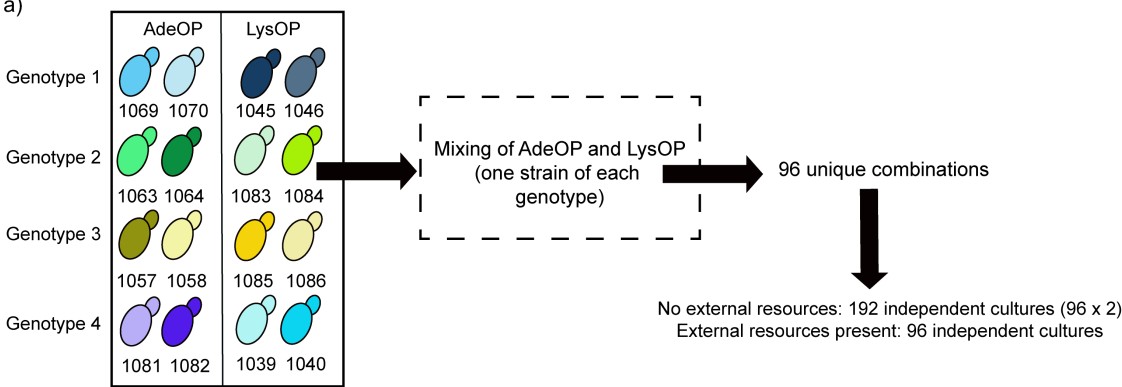

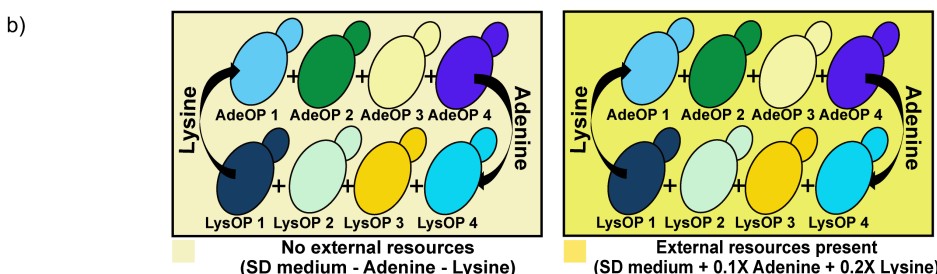

**Fig 1. Overview of the *Saccharomyces cerevisiae* multi-mutualist system, strain types, and experimental environments. (a)** Schematic representation of the engineered strains used in this study and how they were combined to assemble different community types. Two metabolic overproduction backgrounds were used: an adenine-overproducing strain unable to synthesize lysine (AdeOP) and a lysine-overproducing strain unable to synthesize adenine (LysOP). Communities were constructed by pairing four AdeOP strains with four LysOP strains, generating multiple unique cross-feeding consortia **(b)** Experimental media conditions. Communities of *Saccharomyces cerevisiae* overproduction strains growing in media with and without external resources present. The communities were grown in two types of media: no external resources added (light yellow background) and with some external resources added (darker yellow background). Replicate communities growing in these media types were maintained for four weeks.

communities. In the communities that were growing with external resources, we used strains which came from communities where all eight members persisted after four weeks. However, this was not possible for the communities that were evolving in media without external resources due to higher extinctions (Fig 3, S2 Table). In these cases, we had to use communities in which some strains went extinct.

To measure trait evolution, we compared the ancestral strains to their evolved counterparts. We first separated each evolved strain from the others in their community by striking them out on selective plates. We then grew single colony for each strain in YPD for 24 h. This culture was then spot plated and grown in media without external resources to ensure no other strains were present. Each evolved strain was then frozen as above and used as the starting culture for phenotyping assays. Each ancestral genotype was replicated eight times and the evolved genotypes from the two media were replicated 6–8 times.

## Measuring and analyzing strain persistence

The goal behind the analysis of strain persistence dynamics across different media types is to understand how environmental conditions influence mutualist strain survival. We converted the persistence data into a binomial factor, with 1 indicating presence and 0 indicating extinction. The proportion of each strain surviving in all communities was calculated

at each week for different media type. Since proportion data are bounded between 0 and 1, we applied an arcsine square root transformation to stabilize variances and meet the assumptions of parametric testing. We performed a two-way ANOVA in R (v. 2023.06.2 + 561) to test for significant differences in strain persistence across media types, type of over-producer strain (AdeOP & LysOP), and their interaction.

## Starvation resistance measurement

We measured starvation resistance to assess the adaptive responses of yeast strains to amino acid deprivation. We compared differences in starvation resistance between AdeOP and LysOP strains by growing ancestral and evolved strains separately in the adenine and lysine deprived media. To initiate the assay, swabs containing five colonies of each ancestral or evolved strain were grown overnight in YPD. Cultures were then adjusted to a starting density of 0.1 $OD_{600}$ in media lacking adenine and lysine. We counted the number of live cells at 0 h, 48 h and 72 h for the AdeOPs and 0 h, 96 h and 120 h for the LysOPs (based on Vidal et al. [25]). Cells were counted after spreading an aliquot on a YPD plate that was maintained at 30°C. To assess differences in the proportion of surviving colonies among the different media types, we conducted a one-way analysis of variance (ANOVA) using R (v. 2023.06.2 + 561). The proportion of colonies was used as the response variable, and lineage category ("ancestral", "with external resources", and "without external resources") was treated as a fixed factor. For AdeOPs, the change in the proportion of live colonies was examined after 48 h and 72 h whereas for LysOP strains after 96 h and 120 h.

## Overproduction measurements

We chose to measure overproduction to quantify the cost of the mutualism. We grew individual cultures of each evolved and ancestral strain overnight in YPD by using a swab of five colonies from the selective plates. These cultures were then washed with sterile water and diluted to 0.1 $OD_{600}$ in a specific medium. Ancestral and evolved strains of AdeOPs were diluted in SD medium containing 1% lysine (90 mg/L) and LysOPs were grown in SD medium containing 1% adenine (40 mg/L). These cultures were grown individually for 24 h at 30°C on a rotating wheel. After 24 h, we used a small amount of culture to measure the optical density and 800 µl of the culture was centrifuged and then filtered to remove cell debris.

Due to differences in how adenine and lysine are produced, we needed to use two different methods to collect each nutrient. We collected the adenine filtrate by spinning down 800 µl of culture for 1 min at 3700 rpm and then filtering it through a AcroPrep Advance filter plate (Pall Corporation). Collecting lysine was more complicated because lysine is released mainly after the cell death [26]. To collect the lysine filtrate from the LysOP cultures, we took 800 µl of each culture and heat shocked at 100°C for 5 mins, briefly vortexed, and then placed the samples into a dry ice-ethanol bath for 2 mins. We then thawed the samples, and vortexed again for 2 mins. The culture was then spun down for 1 min at 3700 rpm and the sample was filtered using same approach as for AdeOPs. We did not heat shock the adenine overproducers so that we would have a more accurate estimate of the adenine excreted without the addition of adenine from disrupted cells. Thus, separate extraction protocols were used to accurately represent biologically available metabolite levels for each overproducer type.

To measure the amount of adenine or lysine produced by the overproducing ancestral and evolved strains, we used the filtrate to grow a test strain (SY9915, MATa ade2Δ0 ade8Δ0 his3Δ1 leu2Δ0 lys2Δ0 trp1Δ63 ura3Δ0) that cannot produce adenine and lysine. The test strain was prepared by growing overnight in YPD and then diluted in sterile water after washing. We measured adenine overproduction by adding 50 µl of filtrate from AdeOP cultures and 0.1 $OD_{600}$ culture of test strain into 1000 µl of 2X SD media with 1% lysine but lacking adenine. Similarly, to estimate lysine overproduction, we added 100 µl of filtrate from LysOP culture and 0.1 $OD_{600}$ culture of test strain to 1000 µl of 2X SD media with 1% adenine but lacking lysine. The volumes were standardized to 2000 µl by adding sterile water. We took the $OD_{600}$ for the test strain after 24 h and used the corrected OD ($OD_{600}$ corrected = $OD_{600}$ of test strain/ $OD_{600}$ of overproducer strain) for further

analysis. We tested for differences in overproduction between media types with a linear mixed effect model using the lmer function [33] in R, with media type as a fixed effect and the strain ID as a random effect.

### Resource use efficiency and growth rate measurements

We measured resource use efficiency to assess each strain's ability to use traded resources, and thus quantify the benefit gained from mutualism. We define resource use efficiency (RUE) as how much biomass is generated from a limited amount of mutualistic resource. We did this by measuring strain growth when placed alone in medium with a small amount of adenine and lysine. Strains that evolved increased RUE, can turn the same amounts of adenine or lysine into more biomass as compared to the ancestral strain. To measure RUE, we grew individual strains in YPD followed by washing with sterile water and resuspending them in SD medium with 0.05% adenine and lysine. 0.1 $OD_{600}$ cultures of each strain were incubated for 24 h at 30°C. At 24 h, we measured the $OD_{600}$ of the culture. We then compared the yield of ancestral and evolved strains. While the cultures were growing for 24 h we also took $OD_{600}$ measurements at 4 and 6 h to estimate growth rate. We calculated growth rate, $r$, as the number of doublings during exponential growth between 4–6 h:

$$r = \ln(OD_{600}^{t1} / OD_{600}^{t0}) / \ln(2)$$

We tested for differences in RUE and growth between media types with a linear mixed effect model using the lmer function [33] in R, with media type as a fixed effect and the strain ID as a random effect.

## Results

### Media and mutualist type jointly determine strain persistence

With one exception, all the mutualistic communities persisted for four weeks. Strain richness, however, was strongly influenced by media conditions and mutualist type. In obligate conditions (no external resources added), 62% of the communities retained all four LysOP strains while only 8% of communities retained all four AdeOP strains (Fig 2). There was also one community that lost all AdeOP strains. When external resources were available in the media, persistence rates increased as all four AdeOP and LysOP strains survived by the end of four weeks (Fig 2). Furthermore, AdeOP strains started going extinct earlier in the obligate media. Extinction of strains began in two weeks for AdeOPs and in three weeks for LysOPs when in obligate media (Fig 3). A two-way ANOVA revealed a significant main effect of media type ($F_{1,54} = 42.54$, $p < 0.001$). In contrast, the main effect of overproducer type was not significant ($F_{1,16} = 0.00$, $p = 1.00$). Importantly, a significant interaction between media type and overproducer type was detected ($F_{1,54} = 15.78$, $p < 0.001$), indicating that the effect of media on persistence depends on the overproduction classification (mutualist type).

### Starvation resistance remained unchanged across evolutionary environments

Starvation resistance is an important trait that allows strains to survive when resources are limited. Adenine and lysine become highly limiting after each transfer event in media lacking external adenine or lysine but are less limiting in supplemented media. Overall, AdeOPs exhibited lower starvation resistance compared to LysOPs. After 48 h of starvation, only 58% (± 3 SE) of the ancestral AdeOPs survived lysine starvation (Fig 4). AdeOP mutualists that evolved in media without external resources, as well as those evolved in media supplemented with additional adenine and lysine, showed similar levels of starvation resistance at 48 h, with survival rates of 58% (± 7 SE) and 58% (± 4 SE), respectively (Fig 4). At 72 h, 26% (± 1.9 SE) of the ancestral AdeOPs survived. At this time point, mutualists that evolved in media without external resources, exhibited a survival rate of 30% (± 5.5 SE), and those in externally supplemented media showed a survival rate of 30% (± 3 SE). There was no significant effect of media type on the proportion of surviving colonies after 48 h ($F_{2,77} = 0.074$, $p = 0.929$) and 72 h ($F_{2,77} = 0.401$, $p = 0.671$).

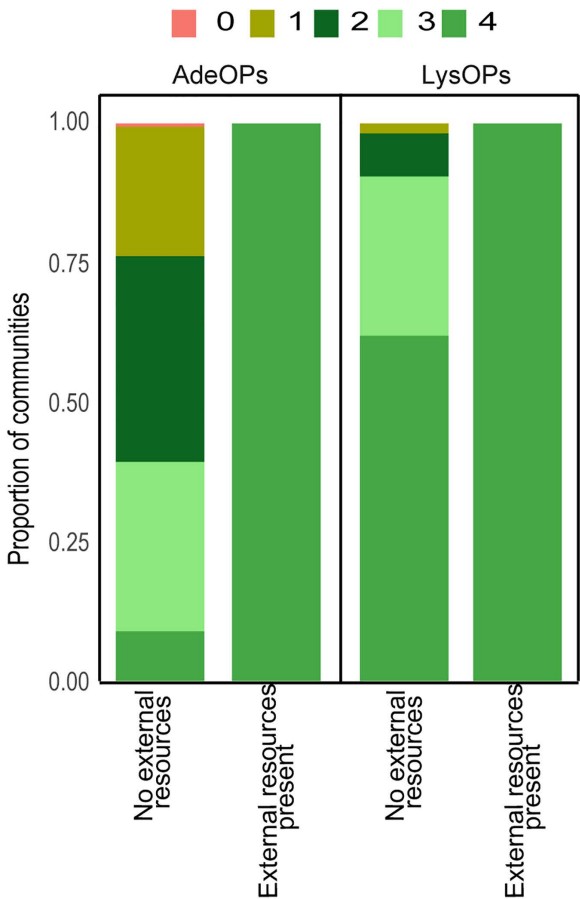

**Fig 2. Plot of proportion of evolving multi-mutualist communities that retained strains of adenine overproducers (AdeOP) and lysine overproducers (LysOP) of *Saccharomyces cerevisiae*.** Communities with no external mutualistic resources in the media lost more strains over time. AdeOP strains were lost more frequently than LysOP strains under obligate and when external nutrients were present in the media. Adding nutrients to the media increased individual strain persistence for both mutualist types.

Similarly, no differences in starvation resistance were observed among LysOPs that evolved in different environments at either 96 or 120 h. At 96 h, 44% (± 2 SE) of the ancestral LysOPs survived adenine starvation. LysOP mutualists that evolved in media without any supplemented resources had a survival rate of 40% (± 3 SE), and those evolved in media containing extra adenine and lysine had a survival rate of 43% (± 3 SE) (Fig 4). At 120 h, 32% (± 1.3 SE) of the ancestral LysOPs survived. At this time point, LysOPs that evolved in media lacking extra resources had a survival rate of 30% (± 2 SE), while those evolved in supplemented media showed a survival rate of 31% (± 2 SE). There was no significant effect of media type on the proportion of surviving colonies after 96 h ($F_{2,79}=0.388$, $p=0.679$) and 120 h ($F_{2,79}=0.18$, $p=0.835$).

**Resource availability increased lysine, but not adenine, overproduction**

The presence and absence of external traded resources as well as mutualist type had significant effects on the evolution of overproduction of adenine and lysine. For AdeOPs, average overproduction did not change significantly from ancestral strains, however, the direction of change was dependent on the media treatments. Overproduction of adenine by evolved strains slightly decreased in obligate conditions but slightly increased when external adenine and lysine were present in the media as compared to the ancestral ($F_{2,74}=4.11$, $p=0.02$) (Fig 5A (a)). As a result, adenine overproduction was

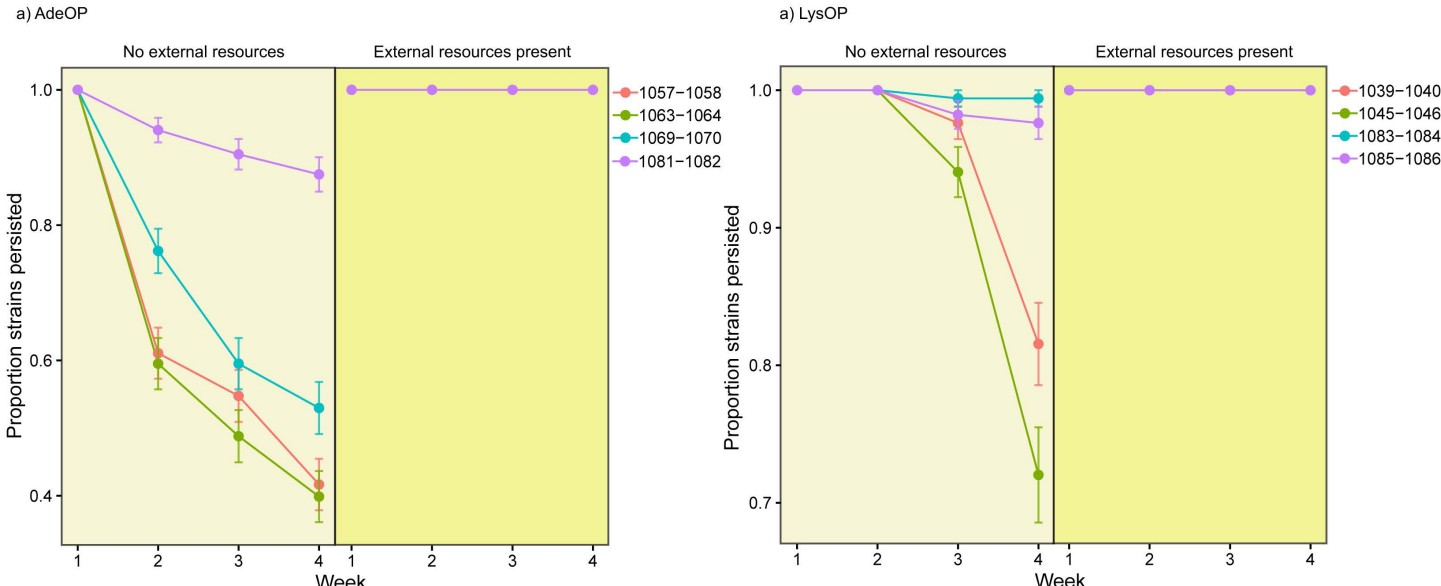

**Fig 3. Individual strain persistence over time of adenine overproducers (AdeOP, left panel) and lysine overproducers (LysOP, right panel) of** *Saccharomyces cerevisiae*. Each colored line represents a distinct genotype within the AdeOP or LysOP guild (strain IDs shown in the legend; full genotypes listed in S1 Table). Error bars indicate the standard error of the proportion of replicate communities in which the strain persisted (calculated as √(p(1−p)/n), where p is the persistence proportion and n is the number of replicate communities). Communities with no external resources in the media lost strains more quickly and more strains over time. This effect was more pronounced for adenine overproducers.

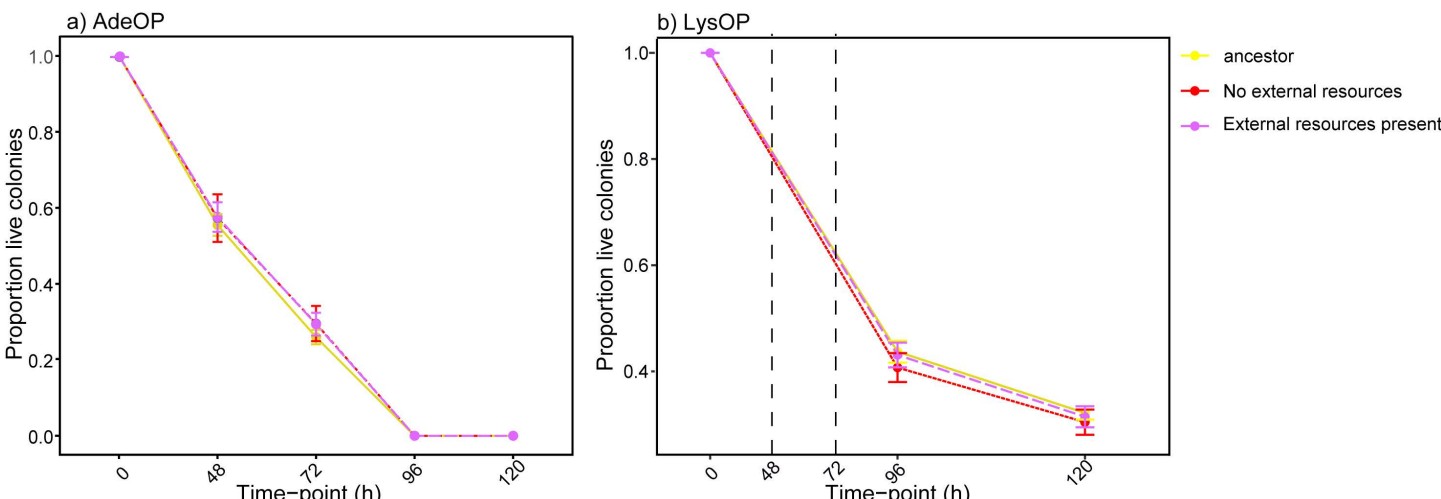

**Fig 4. Starvation resistance of ancestral and evolved** *Saccharomyces cerevisiae* **mutualist types over time. a)** Adenine overproducing (AdeOP) strains perish at a faster rate than **b)** Lysine overproducing strains (LysOP). The dotted lines at 48 and 72 h for LysOP strains are estimates since mortality was first measured at 96 **h.**

significantly different between the obligate and external mutualistic resources treatments. On average, overproduction by adenine mutualists reduced by 1% in the obligate environment, while it increased by 10% in the environment with external resources. Unlike AdeOPs, the lysine mutualists evolved to be the better overproducers of lysine than the ancestral strains

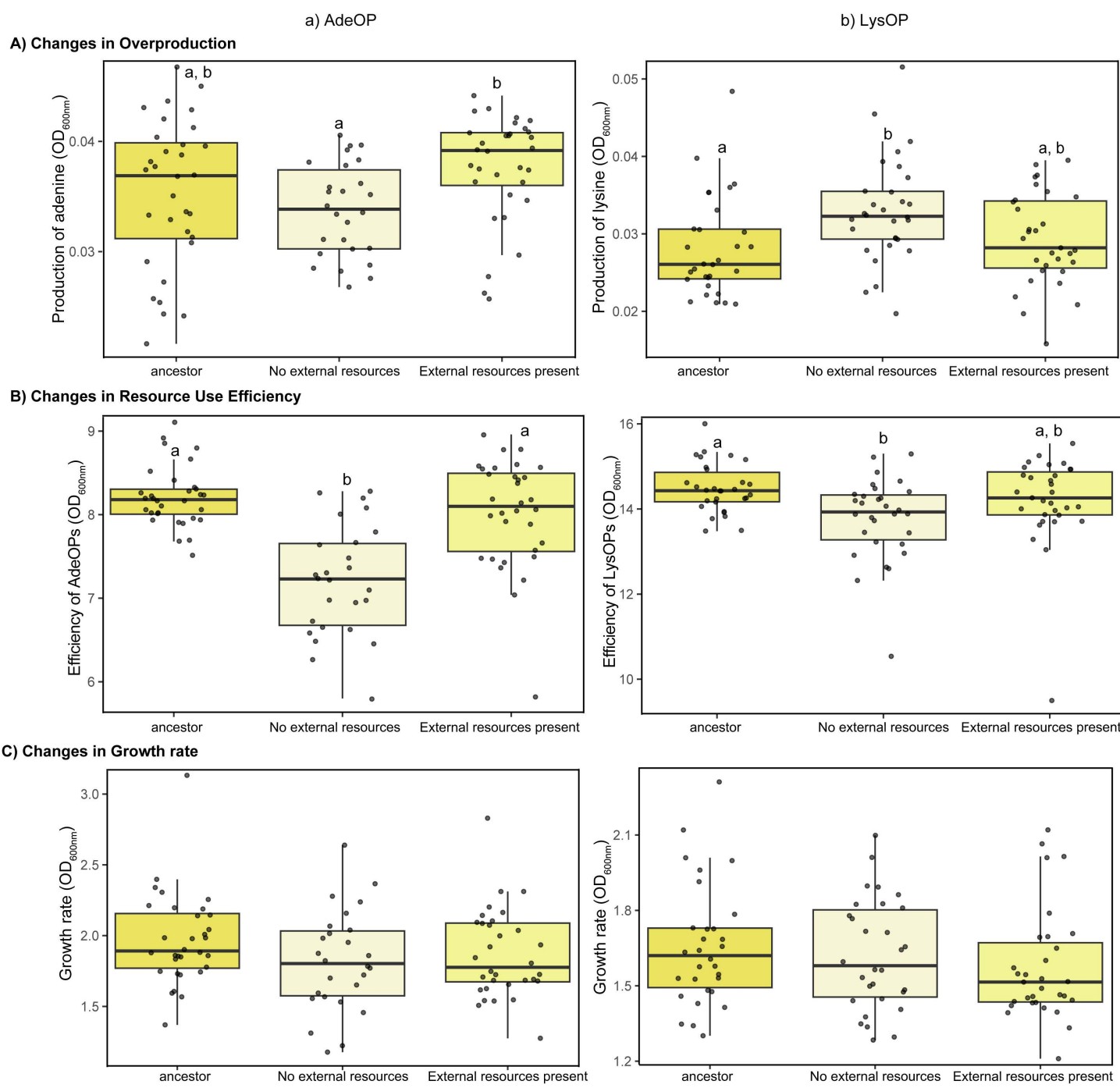

**Fig 5. Measurement of trait evolution.** Changes in **A)** overproduction **B)** resource use efficiency **C)** growth rate in **a)** adenine overproducers (AdeOP) and **b)** lysine overproducers (LysOP) of *Saccharomyces cerevisiae* after 4 weeks of evolution in media with and without external resources present. Statistical analyses were performed using linear mixed-effects models, with significance of fixed effects assessed using F tests (lmerTest package in **R**). Letters denote statistically significant differences at P<0.05. Overproduction and resource use efficiency in both overproducer types changed depending upon the media type in which communities were maintained. Growth rate did not change significantly for either overproduction type.

(Fig 5B (b)). On average, lysine overproduction increased by 4% when external adenine and lysine were present in the media and by 16% in obligate media. Strains evolved in media with external resources overproduced lysine more than the ancestral strains, but less than the strains that evolved in media without external resources present ($F_{2,88}=4.66$, p=0.012).

### Reduced efficiency evolved without growth rate compensation

Adenine mutualists evolved to be less efficient in resource use than ancestral strains and strains that evolved in media with external resources present ($F_{2,74}=20.48$, p<0.0001) (Fig 5B (a)). On average, resource use efficiency decreased by 13% for AdeOPs from the environment with no external mutualistic resources, but only 3% from the environment with external resources. Strains that evolved in media with no external resources were significantly different from the ancestral strains and those evolved in the environment with external traded resources, while the strains that evolved in the environment with external resources did not differ from the ancestral. Overall LysOP strains showed a similar, but smaller change in resource use efficiency compared to AdeOP strains ($F_{2,88}=5.03$, p=0.0085) (Fig 5B (b)). There was a 5% reduction in the resource use efficiency of the lysine mutualists that evolved in the media without external resource, and a 3% reduction for those evolved in the environment with external traded resources. Lysine strains that evolved in no external resources were significantly different from the ancestral, while the strains that evolved in the environment with external traded resources did not differ from the ancestral. There were no significant differences in growth rate for either mutualist type from either evolution environment (Fig 5C (a, b)).

### Discussion

Mutualisms usually involve species from different kingdoms or phylogenetically divergent lineages [2]. As such, it is unlikely that the goods and services provided by each partner species will have the same effect on a mutualist's fitness [17]. This dichotomy in fitness effects can shift the intensity of selection on traits important to facilitating mutualism for each partner. We tested this idea by forming guilds of reproductively isolated strains of brewer's yeast that provided lysine or adenine to their partner guild. Lysine limitation is more costly for yeast physiology than adenine limitation, thus there is an inherent difference on fitness under limited resources. We also examined how the degree of reliance of partner guilds on one another for acquiring these nutrients influenced strain persistence and trait evolution for traits important to the mutualism.

In the synthetic yeast mutualism used in this study, strains differ in their ability to survive without an exchanged resource. Strains that cannot produce their own lysine (AdeOPs) persist in communities less than strains that cannot produce their own adenine (LysOPs) (Fig 2). This dichotomy in the importance of traded resources had cascading effects on individual strain persistence over the course of the experiment. AdeOP strains were more likely to go extinct in communities experiencing an obligate mutualism with LysOP strains (Figs 2–3). All four LysOP strains were retained in over 60% of communities in contrast to all four AdeOP strains being retained in less than 10% of the communities. Our results demonstrate that asymmetries in the fitness effects of traded resources lead to differential persistence of mutualists in these synthetic communities. In this system, lysine scarcity was more detrimental to AdeOP strains than adenine scarcity was to LysOP strains. Consequently, this suggests that the AdeOPs experienced stronger selection pressures and were more prone to extinction when lysine was not available externally. This aligns with the theoretical predictions that mutualists with higher dependency on a traded resource will experience greater vulnerability under obligate conditions [16].

Interestingly, when traded resources were externally supplemented, the disparity in survival between AdeOP and LysOP strains disappeared. All strains persisted in each community over the four-week period (Fig 3). This finding shows how environmental context, specifically the availability of essential nutrients, modulates the intensity of mutualistic dependence and, consequently, mutualist survival. For instance, in plant-mycorrhizal mutualisms, plants rely heavily on fungal partners for phosphorus uptake under low-nutrient conditions but reduce this reliance when soil nutrients are abundant [34–35]. Similarly, legumes form more nodules with nitrogen-fixing rhizobia in nitrogen-deficient soils but downregulate this

mutualism when nitrogen is plentiful [36]. In the yeast multi-mutualist communities, the external supplementation of nutrients relaxed the evolutionary pressure that would otherwise act differentially on strains based on their resource needs. Our results confirm that the strength of mutualistic interactions is deeply context-dependent, with environmental nutrient availability acting as a key driver of mutualist persistence.

In addition to differential persistence, we also observed evolutionary changes in traits linked to mutualism in our experimental system. Resource overproduction, resource use efficiency, and growth traits evolved differently across treatments. Notably, LysOP strains significantly increased lysine overproduction under conditions without external resources, whereas AdeOP strains showed only moderate changes in adenine overproduction (Fig 5A). This suggests that the evolutionary response was greater in the mutualist guild whose partner was more vulnerable to resource limitation. LysOPs evolved increased overproduction under obligate mutualism likely in response to smaller population sizes of their AdeOPs. Increased overproduction could potentially stabilize the interaction by supporting larger populations of their more vulnerable partners. Such compensatory evolution has been observed in nature, where one partner adjusts its contribution to facilitate the mutualistic relationship, especially under asymmetrical dependence. Foster and Wenseleers [37] proposed that reciprocity can be stabilized when more capable or less-dependent partners invest disproportionately in the mutualism, especially when the partner's survival directly affects their own benefit. Harcombe [38] also demonstrated that bacteria evolved to increase the production of essential nutrients when paired with partners that could not survive without them, thereby enhancing community stability. In the yeast multi-mutualist communities, AdeOPs appeared to be more vulnerable to lysine limitation that LysOPs were to adenine limitation. Despite being under strong selective pressure due to their heightened sensitivity to lysine limitation, AdeOPs exhibited a decrease in overproduction and resource use efficiency (Figs 5A and 5B). One potential reason for this is that the surviving AdeOP strains may have evolved strategies to conserve internal resources or reallocate energy away from overproduction toward survival. Consistent with this interpretation, strain-level overproduction assays indicated that all genotypes within each guild evolved in a similar direction, rather than a subset reducing their contribution while relying on others (Fig S1). This pattern suggests a coordinated guild-wide shift in metabolic strategy rather than the emergence of within-guild cheaters. Because AdeOP strains often occurred at lower abundances and were more prone to stochastic extinction during serial transfers, selection may have favored survival-oriented traits over efficient resource production. In this context, maintaining viability under severe resource limitation may have had greater fitness benefits than maximizing the mutualistic exchange. In environments where survival is more important than mutualistic benefit delivery, such as under extreme nutrient limitation, selection may favor metabolic reallocation toward basic maintenance and stress tolerance. Additionally, the serial propagation transfer protocol also reduced traded resources in the environment, especially for the obligate conditions, thus favoring strains that were better at growing faster rather than being more efficient [39]. This often comes at the expense of RUE [40]. A contrasting pattern was observed in pairwise yeast mutualisms, where Wang et al. [41] reported that mutualism promoted higher resource use efficiency and alleviated the tradeoff between growth and efficiency. Our results therefore suggest that in more complex multi-mutualist communities, asymmetries in the importance of exchanged resources can shift selective pressures, favoring survival-oriented strategies in more vulnerable partners rather than increased mutualistic investment. Supporting this interpretation, Vidal et al. [18] showed that many genes for AdeOP strains evolved in pairwise communities showed positive selection for intracellular transport and stress tolerance. Similar kind of adaptive resource conservation has been observed in microbial systems where organisms shift from growth to survival strategies under chronic stress [42–43]. Alternatively, the observed phenotypes could be a result of genetic drift in small populations experiencing frequent bottlenecks. In experimental evolution systems, bottlenecks reduce effective population sizes, allowing random changes in allele frequencies to fix traits irrespective of their fitness consequences [44]. In our yeast mutualism, the frequent 48 h transfer events reduced populations by 90%, which may have contributed to sudden shift in allele frequencies. Additionally, the reduced availability of nutrients when each community is transferred may have selectively favored genotypes reducing overproduction or resource use efficiency and increasing stress tolerance and survival. These

genotypes may have persisted due to lineage competition, even though they are not beneficial for the mutualism in terms of overproduction.

The differences in both persistence and trait evolution show the potential for mutualist guilds to experience divergent evolutionary trajectories based on the relative fitness consequences of traded resources. While members of the guild of AdeOP strains were more prone to extinction, the ones that persisted evolved differently than members of the LysOP guild. Our findings align with previous work showing that ecological dependence and asymmetries in mutualism can lead to divergent outcomes in evolutionary dynamics [16–17]. In mutualistic networks, where multiple species interact, such asymmetries can shape community composition, stability, and coevolutionary outcomes. For example, in plant-mycorrhizal networks, some fungi have many interactions and support a range of host plants, while others are more specialized [45–46]. The stability of such networks may depend on the persistence of these mutualists, which in turn is shaped by their dependence on traded resources.

In conclusion, our study demonstrates that fitness asymmetries in traded mutualistic resources play a key role in shaping both the ecological persistence and evolutionary trajectories of mutualists. Mutualists that suffer greater fitness consequences from lack of traded resources are more likely to go extinct under obligate conditions, while their partners may evolve increased overproduction to compensate for such partner loss. This pattern changes, when traded resources are available from external sources. These results suggest that the balance of mutualistic interactions is sensitive to environmental context and the fitness value of exchanged goods. This work contributes to a growing understanding of how mutualisms evolve and persist, particularly in the context of multi-species interactions and environmental perturbations. Future studies should explore whether similar patterns occur in more complex, natural mutualistic systems and examine the long-term consequences of asymmetric dependency for coevolution and community structure. This system also provides a strong foundation for future modeling work aimed at quantifying how extinction risk and survival-based selection pressures shape evolutionary trajectories in multi-partner mutualisms.

## Supporting information

**S1 Table. Yeast strains used in this study.**
(DOCX)

**S2 Table. Community of yeast mutualists used for phenotyping.** "X" indicates the absence of a strain. In no external resource (obligate) media, only three communities retained all eight strains by the end of the four-week period. Therefore, we also considered communities with fewer than eight strains.
(DOCX)

**S1 Fig. Changes in strain-level overproduction of adenine or lysine in a) adenine overproducers (AdeOP) or b) lysine overproducers (LysOP) of** *Saccharomyces cerevisiae* **after 4 weeks of evolution in media with and without external resources present.**
(TIF)

## Acknowledgments

The authors thank M. Vidal and C. Liu for their valuable discussion.

## Author contributions

**Conceptualization:** Renuka Agarwal, David M. Althoff.

**Data curation:** Renuka Agarwal, Anne E. Curé.

**Funding acquisition:** Kari A. Segraves.

**Methodology:** Kari A. Segraves, David M. Althoff.

**Supervision:** David M. Althoff.

**Writing – original draft:** Renuka Agarwal.

**Writing – review & editing:** Renuka Agarwal, Anne E. Curé, Kari A. Segraves, David M. Althoff.

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
