## [Decision Letter · Decision Letter 0]

21 Oct 2025

Dear Dr. Agarwal,

Thank you for submitting your manuscript to PLOS ONE. After careful consideration, we feel that it has merit but does not fully meet PLOS ONE’s publication criteria as it currently stands. Therefore, we invite you to submit a revised version of the manuscript that addresses the points raised during the review process.

We look forward to receiving your revised manuscript.

Kind regards,

Karthik Raman, Ph.D.

Academic Editor

PLOS ONE

Journal Requirements:

Additional Editor Comments:

The reviews for your manuscript are now in. While the reviewers appreciate the contributions, they have also raised several major concerns, which must be addressed for further consideration of the manuscript for publication. Please take into full consideration the comments by the reviewers and submit a revised version of your manuscript.

Reviewer's Responses to Questions

**Comments to the Author**

1. Is the manuscript technically sound, and do the data support the conclusions?

Reviewer #1: Yes

Reviewer #2: Yes

Reviewer #3: Yes

2. Has the statistical analysis been performed appropriately and rigorously?

Reviewer #1: Yes

Reviewer #2: Yes

Reviewer #3: Yes

3. Have the authors made all data underlying the findings in their manuscript fully available?

Reviewer #1: No

Reviewer #2: No

Reviewer #3: Yes

4. Is the manuscript presented in an intelligible fashion and written in standard English?

Reviewer #1: Yes

Reviewer #2: Yes

Reviewer #3: Yes

Reviewer #1: The article investigates how fitness asymmetries in exchanged resources shape the evolutionary dynamics and persistence of species using synthetic multi-mutualist yeast communities. The authors have used 8 genetically distinct strains of Saccharomyces cerevisiae that either overproduce adenine or lysine but cannot produce the other, the authors simulate mutualism under environments with and without external nutrient supplementation. The study integrates the ecological persistence with trait evolution under different resource contexts, providing direct experimental evidence for how asymmetric fitness consequences can shape mutualism stability. The study is well executed and methodologically rigorous.

Below are my specific comments to improve clarity and presentation. With these minor changes the manuscript is acceptable for publication.

I would recommend the authors to better represent the model being studied. The current figure is confusing and hard to comprehend. My suggestion is to have a better schematic or flow diagram in the Methods section illustrating how the strains were mixed together, why 96 distinct communities were developed, and why this resulted in 288 evolving cultures under various conditions. This would make the design much more understandable to readers.Maintain clear color codes for genetically different strains.

Further, instead of a section named as “strains used in this study” it could be more like “ synthetic model of multi-mutualist communities” . This would emphasize that the work is not only about the strains themselves but about the synthetic mutualism model being tested.

While the use of a growth-based bioassay to estimate adenine and lysine overproduction is valid and functionally relevant, direct quantification (e.g., HPLC, LC-MS, or kit-based metabolite assays) would provide complementary information on exact metabolite levels. I encourage the authors to comment on the choice of indirect quantification based method.

In the "Overproduction measurements" part, Authors have stated that the LysOPs were heat-shocked to release lysine, which is released primarily upon cell death. For the AdeOPs, they just harvested the filtrate. A minor, complementary experiment would be to subject the AdeOPs to the same heat shock and filtration procedure to determine whether it produces more or a greater amount of adenine that could then be utilized to ascertain that adenine is actually released more quickly and more easily than lysine. This could be a minor addition to the procedures. In the case that this is not done, please clarify it in the methodology or discussion

Minor edits:

Line no : 317 (stains to strains )

Ensure consistency in figure references (e.g., “Fig 5a” vs. “Figure 5a”).

A general proofreading pass to correct minor typographical and grammatical errors will improve readability.

Standardize terminology (sometimes “mutualistic goods,” sometimes “traded resources”).

Reviewer #2: Agarwal et al. studied the asymmetric fitness effects of traded resources in a multi-mutualist community. The overall study is excellent, has many strengths, and is written clearly. I only have a few comments (all well within the scope of this paper and without the need for any experiments) that could further improve the study.

I liked the overall findings. Asymmetric fitness effects led to asymmetric trait evolution. Notably, overproduction by lysine producers increased, potentially in a graded response to the expected selection pressure, while adenine overproduction showed very little change.

How does a guild change trait evolution in an asymmetric setting? In my opinion, it is important to discuss how selection might change when moving from a single strain to a combination of strains performing the same function. This is the main idea of this paper. In fact, since you have the data, it would be worthwhile to test whether the results on average trait evolution presented here would differ if considered at the genotype-specific level. I further explain these points below.

Is there any baseline within-guild genotypic variability? Do all genotypes in a guild have the same fitness, with no measurable differences among them, or is it only the case that they all overproduce adenine and underproduce lysine in similar ways, while other differences have not been measured?

Is there a persistent genotype? I would not expect any particular genotype to consistently persist across communities. This can be easily tested.

Do within-guild cheaters evolve? Is overproduction costly? If it is, could some strains within a guild reduce overproduction and act as cheaters, forcing other strains of the same guild to compensate by producing more? Again, this should be straightforward to test by comparing overproduction in specific genotypes, as opposed to comparing only averages from genotype-mixed populations before and after evolution.

Expectation from pairwise combination. Since some results seem to diverge from your previous pairwise study (only one genotype per guild; Wang et al. [39]), could you compare these results against expectations from the pairwise system beyond what is presented in the Discussion?

Ideally, if there is no within-guild variability, then, on average, the species in a multi-mutualist community would feel similar competition and cooperation, and thus their average persistence would match the expectation from the pairwise system. You can easily test this by comparing your pairwise system and multi-mutualist community results side by side. If this comparison is not readily feasible, I understand.

In addition, because of likely stochastic extinction in each transfer window, the selective pressure on a strain with low abundance would be to focus more on survival and could contribute to the reduced resource-use efficiency in AdeOP (a 13% decrease in obligate and 3% in supplemented), while the same did not happen to LysOP (a 5% decrease in obligate and 3% in supplemented). This can also be tested from the available data.

Modeling opportunity for the future. This experimentally grounded system, together with the evolution-specific results from this study, provides an ideal foundation for a computational model that can quantitatively dissect the remaining ecological dynamics and illuminate complex interaction rules beyond net outcomes.

Minor suggestions

Line 308: Increased by 10%??

On average, lysine overproduction increased by 4% when external adenine and lysine were present in the media and by 16% in obligate media. Strains evolved in media without external resources overproduced lysine more than the ancestral strains, but less than the strains which evolved in media with external resources present (F2,88= 4.66, p= 0.012).

It seems inconsistent to me. Should it not instead be, “Strains evolved in media with external resources overproduced lysine more than the ancestral strains, but less than the strains that evolved in media without external resources present”?

In Fig. 3, AdeOP1 represents the arbitrarily assigned first member of the guild, right? If so, the figure must either come from a specific community or represent an average across combinations. Why does AdeOP1 appear significantly different from the others? Why are there no error bars if this represents an average?

In Fig. 4, could you make the x-axis range the same in both?

Figures 5, 6, and 7 could be combined into a single figure. This would help readers view all trait-evolution measurements in one place. Units are missing on all three y-axes. It would be helpful to use the same y-axis range, at least in the growth-rate figure.

Optional: Could you add a panel in Fig. 1 that visually demonstrates the asymmetric effects of adenine and lysine starvation on their dynamics compared to the wild type, coupled with the corresponding adenine and lysine concentrations in the medium? This would make the conceptual asymmetry more intuitive.

Also, please ensure that all figures appear at high resolution in the published version.

Overall, this is an excellent study. I believe the revisions will make the paper more insightful and complete.

Reviewer #3: 1. In methods section, a bit more discussion on the genetic backgrounds of the strains will be helpful.

2. Effective population size during the evolution experiment should be mentioned so that the readers can have an idea about the strength of selection.

3. Why single colonies of individual strains were used for phenotyping after evolution experiment? This would remove

genetic and phenotypic diversity. Only the fixed mutations will be seen and their phenotypes will be measured.

4. 1% lysine - v/v?

5. Is it possible to directly measure lysine and adenine conc. in the filtrates?

6. The statements in lines 303 and 307 seem to be contradicting each other.

7. Line 317 - strains

8. Decreased resource use efficiency in AdeOP strains - I presume that this is for lysine use.

However, we see lysine overproduction in lysOP strain.

Is it possible that the reduction in RUE in AdeOP is caused by lysine overproduction?

9. It is quite puzzling to see reduction in resource use efficiency in LysOP and AdeOP communities in obligate condition.

Why would this happen? It would be good if the authors can discuss this point along with some plausible explanations.

10. Sub-headings in the results could be changed to statements that summarize the results discussed in that section

**Do you want your identity to be public for this peer review?** For information about this choice, including consent withdrawal, please see our Privacy Policy

Reviewer #1: No

Reviewer #2: **Yes:** Aamir Ansari

Reviewer #3: No

---

## [Author Response · Author response to Decision Letter 1]

4 Dec 2025

Comments to Reviewers

We sincerely thank the reviewers for their thoughtful and constructive comments, which have helped us strengthen the manuscript. Below, we provide point-by-point responses and describe the corresponding revisions.

Reviewer #1:

The article investigates how fitness asymmetries in exchanged resources shape the evolutionary dynamics and persistence of species using synthetic multi-mutualist yeast communities. The authors have used 8 genetically distinct strains of Saccharomyces cerevisiae that either overproduce adenine or lysine but cannot produce the other, the authors simulate mutualism under environments with and without external nutrient supplementation. The study integrates the ecological persistence with trait evolution under different resource contexts, providing direct experimental evidence for how asymmetric fitness consequences can shape mutualism stability. The study is well executed and methodologically rigorous.

Below are my specific comments to improve clarity and presentation. With these minor changes the manuscript is acceptable for publication.

Comment 1: I would recommend the authors to better represent the model being studied. The current figure is confusing and hard to comprehend. My suggestion is to have a better schematic or flow diagram in the Methods section illustrating how the strains were mixed together, why 96 distinct communities were developed, and why this resulted in 288 evolving cultures under various conditions. This would make the design much more understandable to readers. Maintain clear color codes for genetically different strains.

Response 1: We understand reviewer’s concern. We have now updated figure 1 for better representation.

Comment 2: Further, instead of a section named as “strains used in this study” it could be more like “synthetic model of multi-mutualist communities”. This would emphasize that the work is not only about the strains themselves but about the synthetic mutualism model being tested.

Response 2: We have changed the “Strains used in this study” to “Synthetic model of multi-mutualist communities”

Comment 3: While the use of a growth-based bioassay to estimate adenine and lysine overproduction is valid and functionally relevant, direct quantification (e.g., HPLC, LC-MS, or kit-based metabolite assays) would provide complementary information on exact metabolite levels. I encourage the authors to comment on the choice of indirect quantification-based method.

Response 3: We thank the reviewer for this insightful comment. We agree that direct quantification methods such as HPLC or LC-MS would yield absolute concentrations of adenine and lysine and provide a more direct measure of metabolite production. However, we used a growth-based bioassay to estimate adenine and lysine overproduction because it provides a functional measure of the metabolites available to mutualistic partners. This indirect approach integrates both metabolite release and biological accessibility, offering an ecologically relevant estimate of the exchangeable resource. Our study focuses on the relative functional differences among treatments rather than absolute metabolite levels. Moreover, this assay is well-validated for Saccharomyces cerevisiae auxotrophs (e.g., Vidal et al. 2020; Shou et al. 2007), allowing us to maintain consistency with prior studies. Additionally, we explored this possibility in the early stages of working with this system, but realized the number of samples we would have to process would make it monetarily prohibitive.

Comment 4: In the "Overproduction measurements" part, Authors have stated that the LysOPs were heat-shocked to release lysine, which is released primarily upon cell death. For the AdeOPs, they just harvested the filtrate. A minor, complementary experiment would be to subject the AdeOPs to the same heat shock and filtration procedure to determine whether it produces more or a greater amount of adenine that could then be utilized to ascertain that adenine is actually released more quickly and more easily than lysine. This could be a minor addition to the procedures. In the case that this is not done, please clarify it in the methodology or discussion

Response 4: We thank the reviewer for this suggestion. We agree that symmetric extraction protocols could improve comparability between overproducer types. However, adenine and lysine differ fundamentally in their cellular release mechanisms. Adenine is secreted continuously into the medium by AdeOP strains during active growth, while lysine is largely retained intracellularly and released upon cell death and lysis. For this reason, we directly harvested the AdeOP filtrate without heat shock, which accurately represents the biologically available adenine under growth conditions. We have now clarified this rationale in the Methods section (line 243-246):

“We did not heat shock the adenine overproducers so that we would have a more accurate estimate of the adenine excreted without the addition of adenine from disrupted cells. Thus, separate extraction protocols were used to accurately represent biologically available metabolite levels for each overproducer type.”

Minor edits:

Comment 5: Line no: 317 (stains to strains )

Response 5: Changed “stains” to “strains” (now in line 337).

Comment 6: Ensure consistency in figure references (e.g., “Fig 5a” vs. “Figure 5a”).

Response 6: We have used “Fig” across entire manuscript.

Comment 7: A general proofreading pass to correct minor typographical and grammatical errors will improve readability.

Response 7: We thank the reviewer for this suggestion. The entire manuscript has been carefully proofread to correct typographical and grammatical errors, ensuring improved clarity and readability.

Comment 8: Standardize terminology (sometimes “mutualistic goods,” sometimes “traded resources”).

Response 8: We have now changed “mutualistic goods” to “traded resources”.

Reviewer #2

Agarwal et al. studied the asymmetric fitness effects of traded resources in a multi-mutualist community. The overall study is excellent, has many strengths, and is written clearly. I only have a few comments (all well within the scope of this paper and without the need for any experiments) that could further improve the study.

Comment 1: I liked the overall findings. Asymmetric fitness effects led to asymmetric trait evolution. Notably, overproduction by lysine producers increased, potentially in a graded response to the expected selection pressure, while adenine overproduction showed very little change.

How does a guild change trait evolution in an asymmetric setting? In my opinion, it is important to discuss how selection might change when moving from a single strain to a combination of strains performing the same function. This is the main idea of this paper. In fact, since you have the data, it would be worthwhile to test whether the results on average trait evolution presented here would differ if considered at the genotype-specific level. I further explain these points below.

Is there any baseline within-guild genotypic variability? Do all genotypes in a guild have the same fitness, with no measurable differences among them, or is it only the case that they all overproduce adenine and underproduce lysine in similar ways, while other differences have not been measured?

Response 1: We appreciate the reviewer’s insightful point regarding potential within-guild genotypic variation. Each strain contributing to a guild was initiated from a single colony, minimizing within-strain genetic variation at the start of the experiment. However, to explicitly evaluate baseline differences among genotypes within the AdeOP guild, we compared ancestral strains grouped by family. Growth rate did not differ significantly among families (F₍3,28₎ = 2.66, p = 0.068), and no pairwise contrasts were significant (p > 0.06). These results indicate that baseline genotypic variability for this trait was limited within the AdeOP guild. Therefore, the guild-level patterns reported in the main text are not driven by pre-existing fitness differences among genotypes, but instead reflect evolutionary responses at the functional-group level. In addition, we examined baseline variation in resource overproduction among AdeOP families. Overproduction differed significantly among families (ANOVA: F₍3,28₎ = 6.57, p = 0.0017), with post-hoc tests indicating that genotype of 1057-1058 exhibited slightly lower overproduction levels compared to the other families (p < 0.012). Although this effect indicates some baseline variation within the AdeOP guild, the magnitude of these differences was modest, and does not alter the overall guild-level conclusions.

We performed the same analysis for the LysOP guild. Ancestral LysOP families did not differ in growth rate (ANOVA: F₍3,26₎ = 0.11, p = 0.95), and no significant differences were detected in overproduction (ANOVA: F₍3,26₎ = 2.07, p = 0.13). We observed a moderate difference in resource use efficiency (ANOVA: F₍3,26₎ = 5.11, p = 0.0065), driven by slightly higher values in strain 1085–1086. Together with the AdeOP analyses, these results show that baseline variation in growth-related performance among genotypes within each guild is limited, while modest differences are present. Therefore, the guild-level trait evolution patterns we report are not the result of strong pre-existing fitness asymmetries among genotypes but instead reflect evolutionary responses occurring at the functional-group level.

Comment 2: Is there a persistent genotype? I would not expect any particular genotype to consistently persist across communities. This can be easily tested.

Response 2: We assume that ‘persistent genotype” means “persistent strain”. We examined persistence at the strain level using the strain-specific data now presented in Fig. 3. While there were differences in the likelihood of persistence among strains within each guild, no strains consistently persisted across all communities. Thus, although persistence varied among strains, these differences did not drive the evolutionary patterns we report, and we do not find evidence for a single “persistent strain” or strain-specific evolutionary advantage. It is possible that genotypes within a strain could evolve and be persistent but testing that was beyond the scope of our study.

Comment 3: Do within-guild cheaters evolve? Is overproduction costly? If it is, could some strains within a guild reduce overproduction and act as cheaters, forcing other strains of the same guild to compensate by producing more? Again, this should be straightforward to test by comparing overproduction in specific genotypes, as opposed to comparing only averages from genotype-mixed populations before and after evolution.

Response 3: We examined overproduction at the level of individual genotypes within each guild to test whether any strain reduced its contribution disproportionately, which would indicate the emergence of within-guild cheaters. The strain-wise overproduction measurements (now shown in Fig. S1) revealed consistent evolutionary shifts across all genotypes within each guild. AdeOP strains showed a similar reduction in overproduction relative to their ancestral values, and LysOP strains showed a similar increase, with no strain deviating strongly from the guild-wide pattern. Therefore, we do not find evidence that any strain evolved a reduced contribution while relying on others to compensate. The observed changes instead reflect a coordinated, guild-wide evolutionary response rather than the evolution of within-guild cheaters (line 411-418).

Comment 4: Expectation from pairwise combination. Since some results seem to diverge from your previous pairwise study (only one genotype per guild; Wang et al. [39]), could you compare these results against expectations from the pairwise system beyond what is presented in the Discussion?

Ideally, if there is no within-guild variability, then, on average, the species in a multi-mutualist community would feel similar competition and cooperation, and thus their average persistence would match the expectation from the pairwise system. You can easily test this by comparing your pairwise system and multi-mutualist community results side by side. If this comparison is not readily feasible, I understand.

Response 4: We agree that comparing to expectations from our previous pairwise system is informative. However, a direct numerical comparison is not appropriate because (i) the pairwise study used a single genotype per guild while our communities contain four per guild, introducing within-guild competition and frequency-dependent effects; (ii) our outcome is persistence across replicate communities (binomial), not continuous growth/efficiency, so the mapping from pairwise dynamics to multi-member persistence is non-linear (iii) the studies were done at different times and with different media, researchers, and other factors that could influence the results. The reviewer’s question is a great one and as part of another study we are going to directly compare the pairwise and 4x4 communities since we are evolving them at the same time.

Comment 5: In addition, because of likely stochastic extinction in each transfer window, the selective pressure on a strain with low abundance would be to focus more on survival and could contribute to the reduced resource-use efficiency in AdeOP (a 13% decrease in obligate and 3% in supplemented), while the same did not happen to LysOP (a 5% decrease in obligate and 3% in supplemented). This can also be tested from the available data.

Modeling opportunity for the future. This experimentally grounded system, together with the evolution-specific results from this study, provides an ideal foundation for a computational model that can quantitatively dissect the remaining ecological dynamics and illuminate complex interaction rules beyond net outcomes.

Response 5: We thank the reviewer for this suggestion. We agree that repeated population bottlenecks may have imposed stronger selection on AdeOP strains to prioritize survival under resource limitation, potentially contributing to the greater reduction in resource-use efficiency observed in AdeOP relative to LysOP strains. We have now added this interpretation to the Discussion, and added a note that this system provides opportunities for future modeling to examine the role of extinction risk and survival-based selection in shaping mutualist evolution.

Line 414-418: Because AdeOP strains often occurred at lower abundances and were more prone to stochastic extinction during serial transfers, selection may have favored survival-oriented traits over efficient resource production. In this context, maintaining viability under severe resource limitation may have had greater fitness benefits than maximizing the mutualistic exchange.

Line 464-466: This system also provides a strong foundation for future modeling work aimed at quantifying how extinction risk and survival-based selection pressures shape evolutionary trajectories in multi-partner mutualisms.

Minor suggestions:

Comment 6: Line 308: Increased by 10%??

Response 6: We agree with the reviewer. We have now made the corrections in line 327-329:

“On average, overproduction by adenine mutualists reduced by 1 % in the obligate environment, while it increased by 10% in the environment with external resources.”

Comment 7: On average, lysine overproduction increased by 4% when external adenine and lysine were present in the media and by 16% in obligate media. Strains evolved in media without external resources overproduced lysine more than the ancestral strains, but less than the strains which evolved in media with external resources present (F2,88= 4.66, p= 0.012).

It seems inconsistent to me. Should it not instead be, “Strains evolved in media with external resources overproduced lysine more than the ancestral strains, but less than the strains that evolved in media without external resources present”?

Response 7: We agree with the comment. We have now corrected the sentence as per suggestion (line 331-334)

Comment 8: In Fig. 3, AdeOP1 represents the arbitrarily assigned first member of the guild, right? If so, the figure must either come from a specific commu

---

## [Decision Letter · Decision Letter 1]

21 Dec 2025

Dear Dr. Agarwal,

Thank you for submitting your manuscript to PLOS ONE. After careful consideration, we feel that it has merit but does not fully meet PLOS ONE’s publication criteria as it currently stands. Therefore, we invite you to submit a revised version of the manuscript that addresses the points raised during the review process.

We look forward to receiving your revised manuscript.

Kind regards,

Karthik Raman, Ph.D.

Academic Editor

PLOS One

Journal Requirements:

Additional Editor Comments:

The manuscript is nearly ready. Pls. Make the minor fixes required by the reviewers and I will accept the manuscript right away. Congratulations on a solid piece of work.

Reviewers' comments:

Reviewer's Responses to Questions

**Comments to the Author**

Reviewer #1: All comments have been addressed

Reviewer #2: All comments have been addressed

Reviewer #3: All comments have been addressed

2. Is the manuscript technically sound, and do the data support the conclusions?

Reviewer #1: Yes

Reviewer #2: Yes

Reviewer #3: Yes

3. Has the statistical analysis been performed appropriately and rigorously?

Reviewer #1: Yes

Reviewer #2: Yes

Reviewer #3: Yes

4. Have the authors made all data underlying the findings in their manuscript fully available?

Reviewer #1: Yes

Reviewer #2: Yes

Reviewer #3: Yes

5. Is the manuscript presented in an intelligible fashion and written in standard English?

Reviewer #1: Yes

Reviewer #2: Yes

Reviewer #3: Yes

Reviewer #1: (No Response)

Reviewer #2: The authors have adequately addressed all of my major and minor concerns, and I recommend the paper for publication. I congratulate the authors on a well-executed study. I have one minor change request that does not require further review.

Figure 4 (Comment 9): I understand and agree with your concern, but my suggestion was purely graphical. Please keep the same x-axis limits to aid visual comparison, even if one curve ends earlier (the remaining region can be left blank). No additional measurements required.

Minor check: Please verify the reported F values around lines 301–303.

Reviewer #3: The authors have addressed all my previous comments satisfactorily.

However, in the caption of figure 5, the authors should mention the name of the statistical test that is being performed.

**Do you want your identity to be public for this peer review?** For information about this choice, including consent withdrawal, please see our Privacy Policy

Reviewer #1: No

Reviewer #2: No

Reviewer #3: No

---

## [Author Response · Author response to Decision Letter 2]

22 Dec 2025

Comments to Reviewers

We thank the reviewers for their careful re-evaluation of our manuscript and for accepting most of our responses from the previous revision. We appreciate the constructive feedback provided in the second round of review. Below, we address the remaining comments and have revised the manuscript accordingly. We believe that these additional clarifications and modifications have further strengthened the clarity and rigor of the study.

Reviewer #1

We thank reviewer for accepting our responses and changes.

Reviewer #2

The authors have adequately addressed all of my major and minor concerns, and I recommend the paper for publication. I congratulate the authors on a well-executed study. I have one minor change request that does not require further review.

Comment 1: Figure 4 (Comment 9): I understand and agree with your concern, but my suggestion was purely graphical. Please keep the same x-axis limits to aid visual comparison, even if one curve ends earlier (the remaining region can be left blank). No additional measurements required.

Response 1: We have revised Figure 4 in accordance with the reviewer’s suggestion by modifying the x-axis to improve visual comparison. Specifically, rather than leaving the terminal time points blank, we have set the proportion to zero, as the proportion of live colonies reaches zero after 72 hours. This provides a clearer and more accurate representation of the observed dynamics across time.

Comment 2: Minor check: Please verify the reported F values around lines 301–303.

Response 2: We thank the reviewer for the comment. After verifying, we have made changes to the lines (now line 295-298 in file with track changes on):

“A two-way ANOVA revealed a significant main effect of media type (F₁,₅₄ = 42.54, p < 0.001). In contrast, the main effect of overproducer type was not significant (F₁,₁₆ = 0.00, p = 1.00). Importantly, a significant interaction between media type and overproducer type was detected (F₁,₅₄ = 15.78, p < 0.001), indicating that the effect of media on persistence depends on the overproduction classification (mutualist type).”

Reviewer #3

Comment 1: The authors have addressed all my previous comments satisfactorily.

However, in the caption of figure 5, the authors should mention the name of the statistical test that is being performed.

Response 1: We have now mentioned the name of statistical test that was performed to make figure 5. We have added the following line in the caption of figure 5 (Line 633-634 in file with track changes on):

“Statistical analyses were performed using linear mixed-effects models, with significance of fixed effects assessed using F tests (lmerTest package in R).”

---

## [Editor Report · Decision Letter 2]

25 Dec 2025

Differences in the fitness effects of traded resources shape traits and persistence in multi-mutualist communities.

PONE-D-25-49040R2

Dear Dr. Agarwal,

We’re pleased to inform you that your manuscript has been judged scientifically suitable for publication and will be formally accepted for publication once it meets all outstanding technical requirements.

Kind regards,

Karthik Raman, Ph.D.

Academic Editor

PLOS One

Additional Editor Comments (optional):

All corrections have been carried out.
---

## [Editor Report · Acceptance letter]

PONE-D-25-49040R2

PLOS One

Dear Dr. Agarwal,

I'm pleased to inform you that your manuscript has been deemed suitable for publication in PLOS One. Congratulations! Your manuscript is now being handed over to our production team.

Kind regards,

on behalf of

Dr. Karthik Raman

Academic Editor

PLOS One